# Search for Novel Lead Inhibitors of Yeast Cytochrome bc1, from Drugbank and COCONUT

**DOI:** 10.3390/molecules26144323

**Published:** 2021-07-16

**Authors:** Ozren Jović, Tomislav Šmuc

**Affiliations:** Ruđer Bošković Institute, Bijenička Cesta 54, 10 000 Zagreb, Croatia; Tomislav.Smuc@irb.hr

**Keywords:** fingerprints, 3D descriptors, similarity, molecular docking, drug repurposing, QM/MM optimization, molecular dynamics

## Abstract

In this work we introduce a novel filtering and molecular modeling pipeline based on a fingerprint and descriptor similarity procedure, coupled with molecular docking and molecular dynamics (MD), to select potential novel quoinone outside inhibitors (QoI) of cytochrome bc1 with the aim of determining the same or different chromophores to usual. The study was carried out using the yeast cytochrome bc1 complex with its docked ligand (stigmatellin), using all the fungicides from FRAC code C3 mode of action, 8617 Drugbank compounds and 401,624 COCONUT compounds. The introduced drug repurposing pipeline consists of compound similarity with C3 fungicides and molecular docking (MD) simulations with final QM/MM binding energy determination, while aiming for potential novel chromophores and perserving at least an amide (R1HN(C=O)R2) or ester functional group of almost all up to date C3 fungicides. 3D descriptors used for a similarity test were based on the 280 most stable Padel descriptors. Hit compounds that passed fingerprint and 3D descriptor similarity condition and had either an amide or an ester group were submitted to docking where they further had to satisfy both Chemscore fitness and specific conformation constraints. This rigorous selection resulted in a very limited number of candidates that were forwarded to MD simulations and QM/MM binding affinity estimations by the ORCA DFT program. In this final step, stringent criteria based on (a) sufficiently high frequency of H-bonds; (b) high interaction energy between protein and ligand through the whole MD trajectory; and (c) high enough QM/MM binding energy scores were applied to further filter candidate inhibitors. This elaborate search pipeline led finaly to four Drugbank synthetic lead compounds (DrugBank) and seven natural (COCONUT database) lead compounds—tentative new inhibitors of cytochrome bc1. These eleven lead compounds were additionally validated through a comparison of MM/PBSA free binding energy for new leads against those obtatined for 19 QoIs.

## 1. Introduction

Cytochrome bc1 is an important anaerobic respiratory chain component, where the electron transfer from substrate ubiquinol to cytochrome c is catalyzed with proton transfer across the membrane of bacteria or fungi [1]. Qo site of cytochrome bc1 is the subject of antifungal agents such as inhibitors such as azoxystrobin, coumoxystrobin, kresoxim-methyl, and fluoxastrobin [2], where inhibitors prevent ubiquinol from binding to the active site [1] (in later text Qo inhibitors, QoI). The basis of such QoI’s mode of action is their higher binding affinity to cytochrome c than ubiquinol [3], and for that reason researches for stronger binding QoI-s have been carried out. So far, ultrapotent site QoI-s have been computationally recommended using the fragment-based drug discovery method [4]. That study was based on finding the strongest binding of the methoxyacrylate mode of action (MOA) antifungal inhibitors when optimizing the fragment attached to fungicide’s pharmacophore with protein side-chain residues in the Qo site. The best hydrophobic aromatic interactions yielded picomolar range Qo site fungicides, which also corresponds to important aromatic–aromatic interactions of famoxadone with the studied side chains [3]. A recent study [5] considers fragment improvements of pyraclostrobin analogs, that contain methoxycarbamate pharmacophore as a MOA chemical group within the same site and target as methoxyacrylate inhibitors. Again, the study reveals optimal fragments having ring–ring hydrophobic interactions between the inhibitor and the protein side-chain residues [5]. Frac-code list [2] contains 21 fungicides altogether as cytochrome bc1 inhibitors at the Qo site labeled as C3 MOA, while listing more than 50 fungicides within the respiration MOA as a general MOA group. The mentioned 21 fungicides at the Qo site cover the following chemical functional groups: methoxyacrylates, methoxyacetamides, methoxycarbamates, oximinoacetates, oximinoacetamides, oxazolidine-diones, dihydrodioxazines, imidazolinones, benzylcarbamates and the most recent tetrazolinone group. Investigations for new lead antifungal compounds while using one very specific chemical functional group within each corresponding article have already been published [4,5]. The searches are necessary to counter the resistance of mutants found in different crop species [2,6], e.g., azoxystrobin and pyraclostrobin are sensitive to G143A mutations [6,7]. Among the mentioned chemical groups, there are, however, either a few or even only a single fungicide for the corresponding group [2], which raises a question of whether there are still other undiscovered pharmacophore functional groups for C3 MOA. One of the novel C3 fungicides is metyltetraprole, reported to be mostly unaffected by mutant pathogens [7], and also stigmatellin, which has a different mode and is unaffected by some mutations [6]. The very recent article considers the use of novel sulfonamides with no positive cross-resistance to eight commercial fungicides (including azoxystrobin) [8]. Whether there are more confounding QoI-s within the known and approved drugs is unknown. For this reason, drug repurposing of the Drugbank compounds for new lead proposals of Qo site inhibitors while searching for new MOA functional groups is still lacking in the literature to the best of our knowledge and should be carried out. Moreover, we consider lead compounds of the newest open natural products database (COCONUT database) [9] to be a very interesting source as well, given that natural compounds have already provided an important role in the discovery of cytochrome bc1 inhibitors [10].

In this work we examined the whole Drugbank (9680 compounds) and COCONUT Database (401,624 compounds) in order to find the subset that was most similar to the C3 group of fungicides, and to allow for new potential pharmacophore groups to be considered in this drug repurposing pipeline. The similarity between the databases and the C3 MOA compounds will be based on molecular fingerprints and stable 3D descriptors of docked conformations. We further decided to restrict selection to compounds with at least an ester or an amide group because those without carbonyl group studies report no significant activity [4,11]. The reason for this is the fact that either one of the two ester oxygen atoms can be involved with the H-bond with amide N-H of the backbone active group in cytochrome bc1. In the case of chain C of yeast *Saccharomyces cerevisiae* (2ibz), it is about Glu-272 residue [1,3]. For the compound to be an active inhibitor, it is necessary either (1) to find a very high scored binding energy hit compound with sufficient H-bond frequency very near the binding site and with any H-bond conformation, or (2) to find a significantly high-scored binding energy compound with a stable and strong conformation precisely containing H-bonds bound to amide N-H Glu272. For Drugbank repurposing, we took into account both approaches ((1) and (2)), but for drug recognition of the COCONUT database, only approach (2) was considered because approach (1) would mean an in-depth search common for repurposing, which is not the case for the COCONUT database containing several hundered thousand compounds, many without stereochemical information, for which no repurposing but only a drug discovery procedure can be considered. The docked compound must have both high docking scores compared to QoI-s and at least one conformation must involve an already precisely defined H-bond. Furthermore, such a compound, when submitted to MD simulation, must have high H-bond statistics and yield relatively strong interactions with the protein when compared to QoI-s. Finally, QM/MM binding energy between hit ligand and protein must be above the margin determined from C3 fungicides; then and only then can such a compound be claimed as an in-silico proposed cytochrome Qo inhibitor lead compound. Determing possible lead compounds with this stringent similarity and modeling filtering pipeline from a large set of synthetic and natural compounds from Drugbank and COCONUT databases is the primary aim of this study. For drug repositioning of the Drugbank database, slightly less stringent criteria will be taken into account in the first part of the pipeline.

While chicken cytochrome bc1 (3TGU) has been already assessed with molecular docking and molecular dynamic simulations [4,5], yeast cytochrome has only been used for experimental studies either as a model system for a variety of plant organisms [6] or in some toxicity and wine-making tests [12,13]. In-silico studies of chicken cytochrome bc1 have been performed but only on several C3 modes of action FRAC fungicides. Mostly, we are speaking of azoxystrobin [4], pyraclostrobin [5], and to some small extent, kresoxim-methyl [11,14], metominostrobin [11], and metyltetraprole [15], while the other 16 QoI-s [2] have not yet been both thoroughly assessed and mutually compared through MD studies to the best of our knowledge. Therefore, describing the H-bond (particularly to Glu-272) of yeast cytochrome bc1 with FRAC QoI-s, and other hydrophobic interactions with MD will be carried out in this study too.

## 2. Results and Discussion

### 2.1. Similarities and Docking

Among the 763 Drugbank compounds that had passed fingerprint similarity condition, 20 compounds have passed descriptor similarity. All details are depicted in Appendix A. When these compounds were submitted to flexible docking, Table 1 displays the results of the highest scored conformation (HSC) and of SHB conformation. Regarding the Drugbank repurposing pipeline, 9 hit compounds out of 20 meet the condition of higher HSC Chemscore fitness than average fungicide HSC and two of them, DB09199 and DB04930, had different H-bonds to those already defined in SHB, but were still considered for MD simulations following the approach (1) due to the attained docking score higher than obtained for the top 1/3 QoIs (which was fenaminstrobin) (Table 1).

Among 863 COCONUT compounds that passed the fingerprint condition, 211 compounds satisfied the 3D descriptors criterion (0.95) and were submitted to flexible docking. Of these 211,62 compounds attained both HSC > 36.2 and at least one SHB conformation was therefore satisfactory and could be further submitted to MD simulations. Appendix A displays the results for these 211 COCONUT compounds. It should be noted that in Appendix A, few compounds were excluded from consideration due to very high Chemscore fitness differences between HSC and SHB conformation. Moreover, a significant number of compounds, although considered for MD runs, have a relatively high gap between the HSC score and the SHB score (Appendix A).

For QoI-s, in 8 out of 21 cases, HSC corresponds to SHB. In 21 cases, stigmatellin does not count because its binding mode does not cover binding with the amide group of Glu-272 residue, but it was submitted to MD runs because it is a protein-docked ligand and assessing its total H-bond frequency and interaction energy is necessary. In the fenamidone case, even after repeated docking with 200 ga runs was performed, no SHB conformation was attained, while fluoxastrobin does not contain a carbonyl group, so these two QoI-s were the only ones which were not submitted to MD runs.

Appendix A displays the number of occurrences with H-bonds including any electronegative ligands’ atoms (or oxygen atom exclusively) and amide HN of Glu-272, where it can be seen that metyltetraprole has the highest number of such conformations among QoI-s, 41 (38) of 100 possible with definition of d(X…H-N) < 3.2 Å and d(X…H) < 2.2 Å. Among hit compounds, CNP0049103 has the highest number of occurrences (80).

The whole 3D descriptor similarity was performed based on solely rigid docking. Therefore, we made additional similarity tests by using top SHB conformations for building their Padel 3D descriptors and using 280 stable descriptors for calculating the similarity between rigid and flexible docking. The median similarity value between rigid-flexible dockings was 0.971, while between all rigid dockings themselves (rigid-rigid) it was 0.984. This means that descriptor values describing the flexible docking conformations are not significantly different to the values obtained for rigidly docked conformations, so the descriptor similarity part of the study does not have to be repeated with flexible docking.

### 2.2. Results of MD Runs

Results of MD runs are displayed in Table 2 where total H-bond count considers all possible H-bonds established between any ligands’ (O, N, Cl or F) or proteins’ (O or N) atoms either being proton donor or acceptor through all 10,001 trajectory points as an average interaction frequency per snapshot. SHB conformations were mostly submitted. From this table it can be seen that among QoI-s, only stigmatellin and coumoxystrobin have a negligible frequency of interactions with amide Glu-272 but they at least have a significant frequency (>10%) of total strong H-bonds (d(X...H) < 2.2 Å, d(O..H-X) < 3.2 Å, in further text ‘Total H2.2’). On the other hand, triclopyricarb has a negligible frequency of Total H2.2 but has more than 10% weak H-bond interactions with amide Glu-272 (X…H–X < 4.0 Å, X…H < 3.0 Å, in further text ‘H3.0 Glu272’). Picoxystrobin and pyrametostrobin have a low frequency of total H2.2 but also more than 10% H3.0 Glu272. All other 15 QoI-s in Table 2 have more than 10% strong H-bonds, almost exclusively with amide Glu-272. It is interesting to note that trifloxystrobin (H3.0 100%) and metyltetraprole (H3.0 100%) constantly hold H-bond interactions with amide Glu-272 during the whole MD simulation. Something very similar might be true for dimoxystrobin, flufenoxystrobin, enoxastrobin, mandestrobin, pyribencarb, and kresoxim-methyl. For that reason, a simple rule for hit compounds can be set that hits must have either more than 10% frequency of total H2.2 or more than 10% of H3.0 Glu-272. For some DB hit compounds, we took into account more conformations, i.e., both SHB conformation and HSC conformation (void of H-bond with Glu-272) if its HSC Gold score (Table 1) was higher than the third top fungicide’s fitness score. Therefore, DB07831 HSC conformation would pass, but SHB would not. The reason for this is that we are testing two approaches with Drugbank compounds, not only SHB conformations (as we do for COCONUT compounds). DB04930 and DB9199 are very high-scored pipeline hits and were therefore exceptionally considered for MD simulations with their conformation having any other H-bond, but the result of the simulation seems to lack both total H2.2 and H3.0 Glu272. DB08439 has conformations of both carbonyl and sulfuryl H-bonds with amide Glu-272, so MD simulations were carried out for both, and both seem to satisfy the set criteria. DB07943, DB08557, DB14668 and DB07181 have very solid H-bond results. DB07181, besides with backbone Glu-272, also attained a very high frequency of H-bonds with its ligand’s amide NH group to Met-139 backbone carbonyl oxygen atom. DB08242 is saved regarding H-bonds with sufficient frequency of H3.0 Glu272. A heat map for Table 2 is depicted in Supplementary information as Appendix A.

Table 3 displays the calculated interaction energies with a Gromacs program during the 1 ns MD simulations. Interaction energy is here defined as the sum of Van der Waals interaction and Coulomb interaction between the ligand and the protein. Here are given average values with standard errors through the whole trajectory (10001 data points considered). *T*-test statistic (<0.01) results show that generally, QoI-s outperform DB hit compounds by comparing their average values (Appendix A). This is expected since most of the hit compounds are not even lead compounds. The simulation of protein–ligand interactions is in water solution under standard dynamic conditions and QoI-s are surely expected to display higher scores. To set a rule for acceptable hit interaction scores, both standard errors of QoI-s and hits must be taken into account. The average QoI interaction energy divided by heavy atom count (HAC) is 7.59 kJ mol^−1^/HA, while the average standard error is 0.52 kJ mol^−1^/HA. Therefore, for QoI-s, the overall (simple) statistic is −7.59 ± 0.52 kJ mol^−1^/HA, while for hit compounds it is 6.72 ± 0.46. The sum of the average standard errors (0.52 and 0.46 kJ mol^−1^/HA) gives 0.98 kJ mol^−1^/HA. Average QoI minus that sum (in absolute terms) gives −6.61 kJ mol^−1^/HA. This will be the cut-off criterion for further pipeline selection to binding QM/MM energy estimations, so all hit compounds with Eint ≤ −6.61 kJ mol^−1^/HA and with sufficient H-bond statistics have passed this pipeline step. These compounds are DB07831, DB07943, DB08439, DB08557, DB14668, and DB07181.

Regarding the MD results with COCONUT hit compounds, for the rules set above, among 62 COCONUT hit compounds, 16 pass to QM/MM. Here, besides interaction energy and H-bond counts, we also took into account RMSD between ligand heavy atoms and backbone protein with elimination of >3 Å. Appendix A displays all the details. Of course, we could inspect more conformations for COCONUT hit compounds with more SHB conformations, but due to the relatively large number of compounds (i.e., 62) put under MD runs in this study, we used only one top-scored SHB conformation. This means that among the 46 failed cases (62 minus 16), some false negatives are possible, e.g., those with a high number of SHB conformations and high RMSD at the same time (Appendix A).

All the mentioned results considered 1 ns trajectories. Experiments with longer MD simulation (10 ns, 40 ns and 100 ns) were carried out too, but results do not significantly differ (in interaction energy and H-bonds) when compared to 1 ns experiments (see Appendix A for detailed results), which is worth noticing.

Scheme 1 and Scheme 2 display a drug repositiong pipeline for DB compounds (Scheme 1) and drug recognition pipeline for COCONUT compounds (Scheme 2). The pipelines are similar, but are different because as already stated, both strong SHB conformation and very strong HSC with HB are considered for Drugbank database, on the other hand only strong SHB is searched for drug recognition of COCONUT compounds. The consequence of this is sligthly different criteria (i.e., more flexible) regarding fingerprints, molecular descriptors and molecular docking are considered for Drugbank repositing workflow. Moreover, the prefiltering step of COCONUT compounds was necessary as mostly these are not drugs but simply natural compounds.

### 2.3. QM/MM Binding Energy

QM/MM Binding energies at different levels of theory have been estimated for (a) rigid docking conformations of all 22 QoI-s (all 21 C3 plus stigmatellin); (b) Gold flexible (HSC) conformations for all 22 QoI-s; for (c) top HSB conformations of 19 QoI-s (all QoI except fluoxastrobin, fenamidone and stigmatellin) and all hit compounds that passed MD simulation step; and for (d) some additional compounds with corresponding HSC or rigid conformations; everything is fully presented in Appendix A. The total number of compounds with calculated QM/MM binding energy equals 77 and, as stated earlier, many of them include more than one conformation evaluated. We present here in Table 4 only the most important results for SHB. We prove that on average SHB conformation binding energies for all C3 fungicides significantly outperform rigid docking (*p* < 0.01) and HSC flexible docking (*p* < 0.05) (Appendix A). The fact that rigid docking is significantly outperformed by flexible docking SHB conformation can be related to initial protein sidechain conformation from crystal PDB with stigmatellin being the binding ligand as it favors the stigmatellin type of binding, since it is contrary to famoxadone C3 type of binding related to most of the analyzed C3 fungicides [1]. Flexible docking enables rearrangement of sidechains (including Glu-272 sidechain) to accommodate C3 fungicides (with different binding than stigmatellin) and establishes an H-bond from carbonyl oxygen to amide group of Glu-272. For rigid docking, there is no SHB conformation for the top-scored QoI conformations. Furthermore, for some rigidly docked conformations, MD simulations have been carried out on some QoI-s (pyraclostrobin, pyrametostrobin, metominostrobin, pyribencarb, famoxadone, enoxastrobin) and the results in most cases are (almost) complete absence of ligands’ formation of H-bonds with any protein’s proton acceptors or donor groups. Binding energies are depicted both with ΔE and with ΔG because estimating ΔG for a higher level of theory with frequencies determined at the lower level of the theory is not very reliable. The difference between two levels for estimating binding energy correction from ΔE and ΔG can highly vary between different cases. When calculating partial hessian, many hindered rotations were treated as vibrations. Therefore, additional scaled correction to the frequencies evaluated at the lower level was estimated from several calculations at a higher level of theory. Still, final ΔG certainly has a higher error margin than ΔE_bind_. Due to the mentioned reasons, we only display ΔG but use ΔE and ligand-size relative Escores for lead compound selections in determining the cut-off level. Average ΔEbind, Escore, and error margins are determined for QoI-s, for both levels of theory. The difference between azoxystrobin’s HSC and SHB conformation is taken as the error margin. It is roughly similar to stigmatellin’s ΔE difference between docked and experimental crystal conformation. The final results reveal that compounds DB07831, DB08557, DB08439, and DB14668 have higher absolute values of ΔEbind and Escore than the average for QoI-s or at least values within the error margin from the average which coincides for both a lower and a higher level of theory (two levels—two different conformations). For some reason, the result of selecting exactly these four Drugbank compounds coincides for both levels with ΔG as these four compounds also either have a higher absolute ΔG or are within the error margin compared to the QoI average, contrary to the other three Drugbank compounds. Therefore, these four mentioned Drugbank compounds have successfully passed all pipeline selection conditions and are finally selected as QoI lead compounds. Among the 16 COCONUT samples (Appendix A), seven of them (Table 4) fulfill the already-stated (QoI average) binding affinity requirements and are selected as lead compounds.

The obtained binding affinity for Pyraclostrobin of −36.2 kcal/mol seems to coincide with the obtained −35.85 kcal/mol on a similar system (on 3TGU target) [5]. The obtained binding affinities could be related to experimental IC50 values on yeast Saccharomyces cerevisiae cytochrome bc1 (wild type) with Azoxystrobin 20 nM, pyraclostrobin 3 nM, and stigmatellin 2.4 nM [6], which seem to roughly correlate to their ΔE and ΔG values, as azoxystrobin shows weaker in-silico binding affinity relative to pyraclostrobin, metyltetraprole and stigmatellin (Table 4). Metyltetraprole and pyraclostrobin also outperform azoxystrobin regarding EC50 values in wild type Zymoseptoria tritici and Pyrenophora teres [15,17]. Bovine cytochrome heart cytochrome bc1 shows higher experimental affinity to pyraclostrobin with relatively similar pKi values of azoxystrobin and kresoxim-methyl, which also approximately correlates with our binding affinity values (Table 4) (azoxystrobin pKi = 8.92, pyraclostrobin pKi = 11.59, kresoxim-methyl pKi = 9.28) [14].

Interaction energies of MD simulations in explicit water solution (Table 3) do not generally correlate with binding affinity estimation in PCM environment (Table 4). However, they at least qualitatively coincide regarding the same compound conformation analysis, where rigidly docked conformations of QoI always seem to have both weaker binding affinity and also smaller interaction energy than their SHB conformation counterparts (Figure 1).

The selected 4 Drugbank and 7 COCONUT lead compounds all obey the rule of five, for COCONUT lead compounds, information can be easily found on the COCONUT web site [18] where each of them is also referenced (under a different name) on the PubChem website, none of them is yet revealed as fungicide or QoI to the best of our knowledge. Scheme 3 displays all selected lead compounds.

Finally, this last QM/MM step might be questioned due to its relatively high computational cost. In that case, the list of hit compounds which passed step 4 can be found in Supplementary information list 1; they could all be potentially considered for future in-vitro testings, in cases where they are afordable, although the complete in-silico pipeline must entail predicted binding energies at the QM/MM level of theory.

Appendix A displays the results of ligand–protein interaction frequencies <3.5 Å for specific protein residues and atoms to study hydrophobic interactions for MD simulations up to 1 ns. C alpha atom (CA) of Gly-143 (CA-Gly143) seems to show on average almost more than one interaction per snapshot for most QoI-s, except for metominostrobin, pyribencarb, and triclopyricarb, but for all studied QoI-s CA-Gly143 > H2.2-Glu272. Only pyribencarb and metyltetraprole H3.0-Glu have a somewhat higher frequency than CA-143. However, only two lead compounds have a higher frequency of H2.2-Glu272 than CA-Gly143, and these are DB08439 and CNP0153569 (H2.2 of 86.25%, Appendix A), which is not the case with any QoI. This proves that forming an H-bond between ligand’s carbonyl group and backbone amide of Glu272 does not have to be followed by an interaction with the C alpha atom of Gly143. DB08439 carries sulfonamide functional group while mutation of Gly-143 to alanine carries a high risk of resistance to the known QoI-s [2,6]. The very recent article recommends the development of sulfonamides as new pesticides due to established not-positive cross-resistance between sulfonamide and eight commercial fungicides [8]. Moreover, lead compounds DB14668 and CNP0083976 have a higher interaction frequency of H3.0-Glu272 than CA-Gly143, similar to metyltetraprole, which is known to be unaffected by Gly143A mutation [7]. Compounds with a low interaction frequency with H2.2-Glu272 (or without such H-bond) seem to show either very high frequency with CE-Met295 (stigmatellin, orysastrobin, coumoxystrobin, DB07831) or with Ar-C1-Phe278(triclopyricarb) or with ArC1-Phe129 (DB08557, picoxystrobin, coumoxystrobin, stigmatellin). These mentioned atoms denote important hydrophobic interactions and pi-pi stacking. Moreover, DB07831 and DB14668 show the record-high number of contacts with C4-ILe125. Interactions with the groups in the similar targets have been described already [4,5,14], but not on Saccharomyces cerevisiae using MD simulations on docked conformations, which is a significantly different system due to initial stigmatellin-type conformation of target sidechains contrary to famoxadone (C3 fungicide) type, especially regarding the Glu-272 glutamate sidechain [1]. All described statistics consider different high RMSDs between ligand-heavy atoms and backbone protein for different cases (Appendix A). Although conformation might vary considerably during the MD simulation, the most QoI-s still seem to retain an H-bond with an amide group of Glu-272 through the whole MD run. This is also true for longer MD simulations (10 ns). For stigmatellin it is uncertain why RMSD is so high since its geometry was submitted to MD after QM/MM optimization with protonated side chain Glu-272 with which it forms an H-bond although in the MD run, such a group is depronated, which definitely might cause significant changes in geometry during the MD run. All other compounds do not form such an H-bond and their RMSD is lower. It must be emphasized that it is difficult to take into account more input conformations for each compound when filtering so many compounds through a pipeline. However, on average, our pipeline approach is acceptable since most QoI-s outperform non-QoI-s regarding the variables from MD simulation (Table 2 and Table 3) and QM/MM binding affinity (Appendix A). Finally, Figure 2 and Figure 3 display visually how the QoI-s and novel lead compounds bind in the active center.

The result of validation steps involving Amber MM/PBSA [19] free binding energy (Appendix A) shows that all 11 selected lead compounds are either within min-max values of 19 QoI-s, or are within one standard deviation of the two lowest-scored QoI-s, although statistically, the average binding energy is slightly in favour of QoI-s (which is expected for the literature established QoI-s). This result regards 1 ns simulation and the similar results and the same conclusion can be established also for 5 ns simulations (Appendix A).

## 3. Materials and Methods

### 3.1. Datasets

From all ca 13,000 Drugbank drugs, we considered structures which contained SDF open structure data file format (i.e., 9680 compounds) (Appendix A). Altogether, 8617 non-missing value Drugbank compounds were taken into account for fingerprint analysis. Azoxystrobin and famoxadone were intentionally excluded because these two already are QoI-s. Regarding fingerprints, we considered all compounds with non-missing values for the following Padel [20] fingerprints: Fingerprinter, Extendedfingerprinter, EStateFingerprinter, MACCSFingerprinter, PubchemFingerprinter, SubstructureFingerprinter. Total number of fingerprints: 4811.

For Drugbank compounds amide or ester functional group filtering was conducted at descriptor similarity filtering, except for the COCONUT Dataset, which contains 401,624 natural compounds (accessed 2 February 2021), prefiltering of compounds was necessary in the start and only compounds satisfying all the following conditions were considered:PSA < 140 and (&)
Molecular weight < 600 Da & 
Number of O atoms (N(O)) > 1 &(1)
2 < N(O) + N(N) < 11 & 
14 < N(C) + N(O) + N(N) < 43 &
−0.4 < AlogP < 5.6

Furthermore, eligible compounds should contain any of the following 8 fragments (SMILES notation): “COC(=O)”, “COC=O”, “C(=O)OC”, “O=COC”, “C(=O)N”, “O=CN”, “NC(=O)”, “NC=O”. Total number of COCONUT compounds that satisfy all mentioned conditiones above are 46,700 compounds and these were submitted to fingerprint analysis, while all other compounds were discarded.

All Padel 3D descriptors were initially considered and used in this study while assessing docked conformations. These 3D descriptors are the following: Autocorrelation3D, CPSA, GravitationalIndex, LengthOverBreadth, MomentOfInertia, PetitjeanShapeIndex, RDF, WHIM. 3D descriptor data were scaled.

### 3.2. Fingerprint Similarity Filtering

Padel molecular fingerprints were calculated for 20 C3 (all C3 except of the recent metyltetraprole) and 1 C8 fungicide ametoctradin as it is also QoI to enable more structural variety of compounds and to open the search for the two mentioned approaches. Jaccard index was calculated between these 21 QoI-s obtaining altogether 210 (i.e., 20 + 19 + ... + 2+1 = 210) Jaccard similarity coefficients within the fungicides labeled as J-Qoin. Jaccard index was also calculated between each inhibitor of these 21 QoI-s in pair with each of 8617 Drugbank compounds, obtaining altogether 21 × 8617 Jaccard similarity coefficients between QoI-s and Drugbank compounds labeled as J-Qo-DB. For each of the 21 QoI-s, the maximum value of the Jaccard similarity coefficient between all other 20 considered QoI-s was taken as a reference obtaining 21 reference J-Qoin values. The first lower quartile of these 21 reference J-Qoin was used for a similarity cut-off for selecting Drugbank compounds similar to QoI-s. For 21 × 8617 J-Qo-DB values, the maximum Jaccard coefficient for each DB compound was selected among the 21 possible ones, obtaining a vector of 8617 maximum Jaccard similarity values for all DB compounds. DB compounds with a maximum value higher than the first quartile of 21 J-Qoin passed this first fingerprint test and were considered for further docking studies; 763 among 8617 passed such a condition and all others were discarded.

The corresponding similarity procedure was carried out also for pre-filtered 46,700 COCONUT compounds with the difference that no ametoctradin, but only 20 C3 fungicides, were used for the first quartile elimination protocol. For 20 × 46,700 J-Qo-DB values, the maximum Jaccard coefficient for each COCONUT compound was selected among the 20 possible, obtaining a vector of 46,700 maximum Jaccard similarity values for all COCONUT compounds. Those COCONUT compounds with a maximum value higher than the first quartile of 20 J-Qoin passed this first fingerprint test and were considered for further docking studies. A total of 863 among 46,700 passed such a condition and all others were discarded. Note that at this stage, more would pass if ametoctradin was taken into account and likely far more if prefiltering of the COCONUT database was not conducted earlier.

### 3.3. Pretreatment of the Protein PDB File

Cytochrome bc1 Complex found in yeast species *Saccharomyces cerevisiae* in complex with Stigmatellin was used as a target PDB for all docking studies, and was downloaded as 2ibz.pdb [21] protein from Research Collaboratory for Structural Bioinformatics (RCSB) Protein Data Bank (PDB). Using GaussView, all water molecules, ions, ligands, and other free molecules were removed; all chains were deleted except Chain C. With Chain C of 2ibz.pdb, extensive and repeated docking studies were carried out for all 20 C3 fungicides and selected hit compounds in both Autodock4 and Gold programs.

### 3.4. Descriptor Similarity Filtering

For 763 Drugbank compounds that passed fingerprint similarity filtering, 3D SDF files were downloaded and converted to a PDB format using an openbabel program with added hydrogens and were submitted to the Autodock4 docking program [22]. Out of 763 compounds, 29 did not contain a 3D SDF file and were evaluated one by one. All results are in Appendix A.

For 863 COCONUT compounds, openbabel generated 3D conformations from flat SDF. Since some compounds contain chiral centers, we carefully assigned stereochemical information for all 863 structures in Appendix A.

Docking was performed on Chain C of 2ibz.pdb using all Autodock4 defaults except the number of ga runs, which was 100 for QoI-s and 20 for Drugbank and COCONUT compounds. Grid cube size was 30 × 30 × 30 Å with 0.375 Å spacings; the cube center was arithmetic mean of the coordinates of Stigmatellin in 2ibz as x = −14.28, y = 38.02, z = −16.80. The whole protein chain was set to be fixed. For 21 QoI-s (21 C3 including metyltetraprole), among 100 ga runs, the top 10 conformations (lowest in energy) were picked out and turned to SDF format and were submitted to a padel program for generation of 3D descriptors. For Drugbank and COCONUT compounds, the same was done but among 20 ga runs, the top 3 conformations were selected.

Among 431 3D descriptors, only 280 descriptors had a stability value of three, defined as the ratio of average and its standard deviation for all 21 Qo fungicides. Three because 99.73% confidence interval covers mean plus or minus three times its standard error and if such interval includes variable’s null value then the variable (descriptor) itself is not considered to be statistically significant (i.e., stable). The cross-validation has shown that in 17 out of 21 QoI cases, at least one conformation out of 10 had a similarity index of 0.95 among these 280 3D descriptors. Cross-validation similarity index for a tested cross-validated conformation (left-out conformation) is defined as the ratio of those stable 3D descriptors that are within two standard deviations from the average descriptor value of the training fungicide set and all 280 stable descriptors considered, i.e., a similarity of 0.95 means 0.95 × 280 = 266 descriptors (at least) for considered compounds that have values within 2 standard deviations from the mean values of 21 (C3) QoI values.

For each Drugbank and COCONUT compound, the compound is determined to be similar to QoI-s if any of the three conformations considered had a similarity of 0.95, and if the compound has at least one X-C=O (X=O or N but not OH) functional group. Then, the compound has passed to molecular docking studies. This is our pipeline procedure.

An additional internal test was carried out with azoxystrobin (as DB07401) included in the set along with the rest 763 compounds, to see whether or not docking with only 20 ga runs would yield 3D descriptors similar enough to those with 100 ga runs of azoxystrobin, and the result was that azoxystrobin passed the pipeline with 0.971 similarity (>0.95).

To test for robustness only for the Drugbank dataset, the additional selection procedure was carried out, this time, instead of metyltetraprole, stigmatellin was used with the rest 20 QoI-s for creating the stable descriptor set. With the stability of three, the same 3D descriptor set of 280 variables was used, and the cross-validation has shown that in 18 out of 21 cases, at least one conformation out of 10 had a similarity index of 0.95 among these 280 3D descriptors. The new set of Drugbank hit compounds with 0.95 similarity was selected and was shown to be larger than the former set. All compounds in the new set that would be selected in the former set with a similarity of 0.93 were also recognized as hit compounds. This was not done for the COCONUT set.

Besides 0.95 similarity criteria (and its robust addition), some additional pipeline routes were considered for drug repurposing of the Drugbank database, but it did not finally lead to any new lead compund (for more details see Supplementary information note 1).

### 3.5. Molecular Docking

Autodock4 served for all rigid dockings (especially regarding former Section 3.4) while Gold was used for all flexible docking simulations. Please note that this was done in order to clearly discriminate rigid docking from flexible docking, although for rigid docking, the Gold program can also be used instead of Autodock4, as additional rigid docking tests in Gold have shown almost the same results concerning both descriptor similarity and RMSD.

#### 3.5.1. Flexible Docking

All 21 C3 fungicides and stigmatellin (i.e., 22 QoI-s), along with all hit compounds (of both datasets), were used for Gold docking simulations [23]. The same grid center coordinates were used as in Autodock4 and all the atoms were selected within a 30 Å distance using the detect cavity option. All H-bond donors/acceptors were forced to be treated as solvent-accessible. A “chemscore_kinase” template was used and loaded and the scoring function was ChemScore. The genetic algorithm settings were user-defined with the default initial values. Opposite to the procedure where rigid protein was used in simulations, all Gold runs were carried out using the following 10 flexible aminoacid sidechain residues in Chain C 2ibz.pdb: Ile-125, Phe-129, Tyr-132, Met-139, Val-146, Ile-147, Ile-269, Glu-272, Leu-275, and Met-295. The Glu272 sidechain COO^–^ group was protonated (COOH) in this Gold docking, since water molecules form an H-bond with a carboxylate sidechain of Glu-272 and serve also as proton donors to fungicide famoxadone [1]. The flexible sidechains are mandatory for C3 fungicides, especially in this case, because the beginning conformation is stigmatellin conformation, and it is already known that the sidechain position of Glu-272 in the case of stigmatellin is different than in the case of famoxadone [1,3]. For both fungicides and hit compounds, the experiments were performed using 100 ga runs (without earlier termination). Gold Chemscore fitness above the average of QoI was used as a pipeline selection cut-off for hits priorly selected with 0.95 (or 0.93) similarity. Finally, hit compounds above the cut-off were further tested if at least one out of the 100 conformations contain a possible H-bond between ligand’s carbonyl (C=O) oxygen atom and proton donor amide group of Glu-272 residue with conditions of d(O…H-N) < 4.0 Å and d(O…H) < 3.0 Å. If it contains it, the conformation with such an H-bond of the highest possible Chemscore fitness (if more H-bond conformations to Glu272 were obtained) was taken for further MD-simulation and labeled as a specific H-bond (SHB) conformation (“SHB conformation” in further text). More precisely, firstly, the top-scored conformation with d(O…H-N) < 3.2 Å and d(O…H) < 2.2 Å was searched and if no such conformation was found then the search considered the conformation with d(O…H-N) < 3.6 Å and d(O…H) < 2.6 Å and finally, the search took into account the last possibility of the top-scored H-bond with criterion d(O…H-N) < 4.0 Å and d(O…H) < 3.0 Å. For fenamidone and flufenoxystrobin, additional docking with 200 ga runs and with 20 Å distance atom selection was utilized.

#### 3.5.2. Rigid Docking

Rigid docking was performed using Autodock4 on Chain C of 2ibz.pdb using all QoI-s and some Drugbank hit compounds with the same conditions mentioned in the previous section but this time all docking experiments were carried out with 100 ga runs. Minimum energy was used as input for further QM/MM optimization runs for QoI-s. It is very important to note that repeated docking experiments were conducted to see RMSD for the same compound between minimum energy conformations among 100 ga runs. Repeated runs were carried out for stigmatellin, which had its crystal conformation in RCSB data bank as 2ibz.pdb. The average obtained RMSD for seven dockings (6 using Gold and one using Adt) between docked and experimental geometry of stigmatellin within the target protein was 1.60 Å with a standard deviation of 0.40 Å.

### 3.6. Molecular Dynamics

Molecular dynamics was carried out in the Gromacs program [24] using the detailed instructions in the Supplementary_MD_instructions folder (with all the input files), and following the script ‘main’. Protein-ligand Gold (mostly SHB) docked conformations were taken as input coordinates for MD simulation steps. The first step was to generate a topology for protein and ligand and then to build the complex. The protein-ligand system size of 4.731 nm × 8.057 × 5.721 nm cube box of edge length 10.792 nm was added, filled with water molecules, and neutralized with chlorine ions (here Glu-272 sidechain was deprotonated). Then, energy minimization was carried out, which was followed by NVT equilibration and then NPT equilibration; finally, MD runs were carried out. Equilibration runs lasted 100 ps for NVT and 100 ps for NPT, MD simulation lasted for 1 ns, with one step period of 2 fs. Every 100 fs coordinates and energies were saved, obtaining altogether 10 001 simulation steps (starting from 0 to 1 ns). A Charmm36 + TIP3P force field was used. With the MD simulation finished, a PDB file for the whole trajectory was created and total H-bonds between ligand proton donor/acceptor atoms (O, N, F, and Cl) and between protein proton donor/acceptor atoms (O and N) were searched with a criterion of satisfying both d(X...H) < 2.2 Å and d(X...H−X) < 3.2 Å. Besides total H-bond count, specific H-bonds were counted between the ligand and amide groups of Glu-272 with more H-bond distance definitions. Specific contacts (and or H-bonds) between ligand and residue atoms of Ile-125, Phe-129, Tyr-132, Gly-143, Phe-278, Tyr-279, and Met-295 residues were also counted [4]. Finally, the average interaction energy between ligand and protein was calculated through the whole trajectory, as the sum of Coulomb and Van der Waals contributions and were used for pipeline evaluations.

Besides 1 ns MD simulation runs conducted for all QoI and Drugbank hit compounds, 10 ns MD simulations were carried out with 10 ps snapshot step (i.e., altogether 1000 snapshots). In addition, for nine selected ligands (six QoIs and three hit ligands), 40 ns experiments with 40 ps snapshot step and 100 ns experiments with step period of 100 ps were conducted [25]. The results of prolonged simulations were used to validate our choice to estimate binding affinities from short (1 ns and 10 ns) simulation runs.

### 3.7. QM/MM Docking Optimization

For QoI-s and hit ligands (that passed all prior steps), SHB conformations and in some cases top-scored Autodock4 conformations and/or top Gold scored conformations were submitted to ORCA version 4.2.0 [26] DFT QM/MM docking optimization.

Chain C of 2ibz void of any other ligands, water molecules, and hemes was submitted to a VMD program where Protein Structure Files (PSF) and PDB files were constructed for the ORCA input. The ORCAFF.prms file for the protein was made using a Charmm force field. The ligand charges were initially created with ‘makeff –PBE’ obtaining Chelpg charges while retaining all the ligand coordinates from the Autodock4 docking log (DLG) output file [27] in the case of rigid Autodock4 docking. In the case of the flexible Gold docking, both the protein and ligand coordinates were extracted from Gold and converted using Gaussian and our scripts (Supplementary_file_from_Gold_to_MD_and_orca.docx) to match ORCA input files (PDB and ORCAFF.PRMS for protein and XYZ and ORCAFF.PRMS for ligand). A ligand force field was later merged with the protein. The optimization runs were set with the ORCA defaults, employing the keyword “Opt”. The Polarizable Continuum Model (PCM) was used in all the calculations with water used as solvent. After the convergence of the ligand–protein complex, the sole ligand and the sole protein were optimized with initial input coordinates the same as those obtained for the converged protein-ligand structure. The whole ligands were taken as QM active. For the protein, the following amino acid side chains were used as QM: Ile-125, Phe-129, Tyr-132, Met-139, Val-146, Ile-147, Leu-150, Ile-269, Glu-272, Leu-275, Tyr-279, Leu-282, Met-295, Ile-299. All these mentioned were also taken as active with additional active amino acid side chains: Thr-122, Val-270, Pro-271, Phe-278, Phe-296. The Glu272 sidechain COO^–^ group was protonated (COOH) and ligands were treated as neutrally charged species so that the complex was QM neutrally charged (only in one case the sidechain was deprotonated due to the positively charged amino ligand group of DB07181). Functional used was B97-D3 [28,29], the basis set was 3-21G. After geometry convergence, the final single point energy was calculated using a larger basis set (def2-SVP def2/J and Grid6) and using a geometry counterpoise correction “GCP(DFT/SVP)”. Binding electronic energy (ΔE) was calculated as a difference between complex single point energy and the sum of sole protein and sole ligand single point energies.

After obtaining binding energies at B97-D3/def2-SVP//B97-D3/3-21G, the procedure was in the most important cases repeated at B97-D3/def2-SVP//B97-D3/def2-SVP, starting at priorly optimized coordinates (at lower level). Binding energy was calculated again.

In the most important cases, harmonic frequency calculations were calculated for B97-D3/3-21G optimized geometries to estimate Gibbs free binding energy by taking into account vibrational zero-point energies, thermal enthalpy correction at 298 K, and all entropy corrections. Only in few cases were very expensive frequency calculations carried out at B97-D3/def2-SVP optimized geometries.

### 3.8. Amber MM/PBSA Binding Energy

All 19 considered QoI-s and 11 passed lead compounds were submitted with their “optimal” Gold docking conformation (mostly containing SHB conformation) to MM/PBSA [30] free binding energy calculation in Amber program. Firstly, the protein’s coordinate and topology files were created from ORCA PDB file using Ref. [31] with implicit solvent model at pH = 7.4 and ionic strength = 0. With amber script and tleap program, the cube system was formed with its PRMTOP and INPCRD files, containing: protein, ligand, water molecules and (six) chlorine counter ions (to neutralize the system). Using ante-MMPBSA.py script PRMTOP files for complex, protein and ligands were created. The system was submitted to minimization step (maxcyc = 10,000), then to NVT equilibration followed by NPT equilibration runs (each 100 ps long with 1 fs step). Optimized isokinetic Nose-Hoover chain ensemble was used (ntt = 9) in equilibration runs [32]. After equilibration at 300 K and 1 bar, MD production runs were carried out for 1 ns (each step of 1 fs). A total of 200 equidistant conformations were sampled (each mutually 5 ps distant) and submitted to MM/PBSA energy evaluation [33]. All scripts can be found in supplementary MD instructions Amber_mmpbsa_in folder.

In addition to 1 ns MD simulations, we performed 5 ns MD simulations for 25/30 of the QoI and lead compounds, using 25 ps as a snaphot step (i.e., altogether 200 snapshots)”.

## 4. Conclusions

We carried out extensive in-silico drug recognition of Drugbank and COCONUT databases and found altogether 11 new compounds with potentially strong binding to cytochrome bc1 Qo site. This binding might inhibit electron transfer from ubiquinol to cytochrome c. Among the 11 identified compounds, one of them (CNP0361420) contains the benzyl-carbamate chemical functional group, which is one of FRAC C3 MOA group [2], other compounds contain not-yet-established MOA groups specific to the Qo site, but the sulfonamide functional group (of identified DB08439) is already recognized as having an antifungal mode of action [8]. Lead compounds picked out by the pipeline were additionally successfully validated by Amber free binding energy calculation (MM/PBSA energy). Novel studies can test our in-silico identified compounds in vitro and in vivo for antifungal activity and can probably carry out candidate optimization of lead compounds’ fragments and/or experimentally test our compounds on mutated species.

## Data Availability

The data presented in this study are available in Appendix A.

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
