# Peer review of "Search for Novel Lead Inhibitors of Yeast Cytochrome bc1, from Drugbank and COCONUT"

_molecules, 2021, doi:10.3390/molecules26144323_

Round 1

Reviewer 1 Report

This paper describes an approach to select lead compounds for cytochrome bc1 inhibitors as fungicides. To select the compounds, they used several steps including docking simulation, molecular dynamics, and QM/MM calculations. Several known fungicides are used as control and the selected compounds are scored with almost same levels of control compounds.

The manuscript is well prepared and the experiments and results are scientifically sounds.

Therefore, this reviewer concludes that this manuscript is acceptable for publication, but would request to consider minor points below. 

1. The authors used two databases, Drugbank and Coconut database, and schemes applied to select compounds are slightly different. The reason should be described on this point, are the necessary information different in each database? If the information is the same, how do authors apply a scheme to a combined (or mixed) database of DB and CNP?

 2. Line 127, just typo BD07174 -> DB07174 

Author Response

Reviewer 1. This paper describes an approach to select lead compounds for cytochrome bc1 inhibitors as fungicides. To select the compounds, they used several steps including docking simulation, molecular dynamics, and QM/MM calcluations. Sveral known fungicides are used as control and the selected compounds are scored with almost same levels of control compounds.

The manuscript is well prepared and the experiments and results arescientifically sounds.

Thefore, this reviewer concludes that this manuscript is accpetable for publication, but would request to consider minor points below.

  1. The authors used two databases, Drugbank and Coconut database, and schemes applied to selected compounds are slightly different. The reason should be described on this point, are the necessary information different in each database? If the information is the same, how do authors apply a scheme to a combined (or mixed) database of DB and CNP?

A: Two schemes have to be slightly different due to the following reasons: I) As stated in the introduction section for the Drugbank database drug repurposing (repositioning) has been carried out, while in the case of the Coconut database drug recognition (discovery) has been performed. II) Drug repurposing of the Drugbank database considered both a) SHB conformation and b) strong HSC, while recognition of Coconut compounds considered only SHB conformation (see introduction section). III) Finally, necessary information is different due to the fact that the Coconut database does not contain stereochemical information and at the same time it contains many compounds with a very low fraction of approved or investigational drugs. This means that by definition Coconut database simply cannot be drug repurposed since we are not dealing with drugs, especially not if the chiral centers on some compounds are not known. The Drugbank database on the other hand is well defined. In the case of the Coconut database, the drug discovery approach had to be carried out, where the interest can only be to find strong SHB conformation. For the mentioned reasons, more stringent criteria (for fingerprints, descriptors, and docked conformations) had to be set in the first few steps of the pipeline. So, our answer is that the content, in terms of compounds and associated information, is not the same in these two databases  - which is the reason we have, originally, used two different approaches.

  1. Line 127, just typo BD07174 ->DB07174.

A: We followed the reviewer's suggestion and made necessary changes. We thank the reviewer for the instructive review.

Reviewer 2 Report

Over all, this manuscritp was well-designed and easy to understand. In this work, the authors applied classical drug design approaches for finding potential inhibitors of  Yeast Cytochrome bc1. Just few concerns to the authors and then I think this manuscript could be accepted for publishing in Molecule Journal.

  1. Figures 2 and 3 should be re-depicted for better representation.
  2. QMMM and MMPBSA should be replaced by QM/MM and MM/PBSA.
  3. Some important references of MM/PBSA free energy calculations should be cited.

Author Response

Reviewer 2. Over all, this manuscript was well-designed and easy to understand. In this work, the authors applied classical drug design approaches for finding potential inhibitors of Yeast Cytochrome bc1. Just few concerns to the authors and then I think this manuscript could be accepted for publishing in Molecule Journal.

  1. Figures 2 and 3 should be re-depicted for better representation.

A: We followed the reviewer's suggestion and made appropriate changes.

  1. QMMM and MMPBSA should be replaced by QM/MM and MM/PBSA.

A: We followed the reviewer's suggestion and made appropriate changes.

  1. Some important references of MM/PBSA free energy calculation should be cited.

A: We followed the reviewer's suggestion and added three more references.

We thank the reviewer for the constructive review.

Reviewer 3 Report

The authors of the paper introduced (according to them) a novel method for prioritising candidates for novel outside inhibitors of cytochrome bc1. They searched for candidates from the pool of compounds of Drug Bank and COCONUT, a composite database of natural products.  They assembled the pipeline of different methods, which are shown in Schemes 1 and 2. The authors finally proposed four new inhibitors from the drug bank and seven from the COCONUT database.

1.) The authors used a COCONUT database composed of many databases. I wonder how they considered the stereochemical information that is missing for numerous compounds in this database.

2.) Very unusual analysis of MD trajectory in relation to HB. Using % of time HB is present between HBA and HBD is more informative than numbers. Perhaps a heat map would be more informative.

3.) I can't understand sentences like this: "The C-alpha atom of Gly-143 347 (CA -Gly143) appears to show nearly or more than ten thousand interactions for most 348 QoI-s, except for metominostrobin, pyribencarb, and triclopyricarb, but for all 349 QoI-s studied CA -Gly143 H2.2-Glu272." How can one atom (CA of Gly 143) make more than ten thousand interactions with the low molecular weight ligand?

4.) Figures 3. and 4. showing the complex between receptor and different ligands are very indistinguishable and not informative, at least the ligand should be coloured differently than the receptor atoms. The authors should improve the representation of the complexes by hiding the aliphatic atoms.

5.) Scheme 3. The authors should show the molecular structures and not cut’n’paste.

 6.) MD trajectories (1 ns) are clearly too short to obtain relevant information, for analysis 1000 snapshots during 100 ns would be optimal.

7.) MM/PBSA, QM /MM etc. are correct abbreviations for the methods.

8.) The pipeline is too complicated, I wonder if it makes sense to dock with GOLD and AD 4.2. Is the last step QM /MM necessary?

The topic addressed in this paper is potentially relevant and could be accepted for publication after clarifying many issues.

Author Response

Reviewer 3. The authors of the paper introduced (according to them) a novel method for prioritising candidates for novel outside inhibitors of cytochrome bc1. They searched for candidates from the pool of compounds of Drug Bank and COCONUT, a composite database of natural compounds. They assembled the pipeline of different methods, which are shown in Schemes 1 and 2. The authors finally proposed four new inhibitors from the drug bank and seven from the COCONUT database.

1) The authors used a COCONUT database composed of many databases. I wonder how they considered the stereochemical information that is missing for numerous compounds in this database.

A: As already replied to reviewer 1, two different approaches and schemes are applied regarding Drugbank and COCONUT. For Drugbank we used drug repurposing, while for COCONUT we applied drug recognition (i.e. drug discovery). First, we prefiltered COCONUT and reduced the number of compounds from 401 thousand to 46700 compounds. Then we used the fingerprints based filter with which we further reduced to 863 compounds. For these two steps - prefiltering and fingerprint step 3D i.e. stereochemical) information is not necessary. For the next 3D descriptor step we built 3D structures from flat SDF files using openbabel program. And the reviewer is right since before we haven't put the 3D output regarding the chiral centers for many compounds. In our new version of the paper, we checked and assigned all existing chiral centers (if present) for each of 863 compounds in step 2 for the COCONUT database in Supplementary table S11 (with R or S or with N in case if there are no chiral centers or with more R/S in case of more chiral centers). Accordingly, one cannot drug repurpose the database that contains low fractions of approved or investigational drugs, mostly containing natural compounds, which are not drugs. In that case, stereochemical information is not obligatory but must be built and assigned, which we now provide in our revised version of the manuscript in Table S11.

2) Very unusual analysis of MD trajectory in relation to HB. Using % of time HB is present between HBA and HBD is more informative than numbers. Perhaps a heat map would be more informative.

A: We followed the reviewer's suggestion and made appropriate changes. In particular, we maintained table 2 but used percentages of time instead of the total frequency of interactions. We additionally plotted a heat map and placed it in the novel Supplementary Figure S1 describing these percentages between different samples.  

3) I can't understand sentences like this: "The C-alpha atome of Gly-143 (CA-Gly143) appears to show nearly or more than ten thousand interactions for most QoI-s, except for metominostrobin, pyribencarb, and triclopyricarb, but for all QoI-s studied CA-Gly143 H2.2-Glu272". How can one atom (CA of Gly 143) make more than ten thousand interactions with the low molecular weight ligand?

A: We thank the reviewer for spotting this shortfall.

 Ten thousand interactions refferred to the interactions over the whole trajectory – which indeed was not clearly stated in the paper. This meant that on average there was more than one interaction (between CA-Gly143 and any ligand atom) per snapshot within 3.5 Å. We carefully checked all sentences and made appropriate changes.

This remark also warned us that Figure 5 is missing in our submitted manuscript, although we were referring to it in the main body text. Now we added Supplementary Table S8

4) Figures 3 and 4 showing the complex between receptor and different ligands are very indistinguishable and not informative, at least the ligand should be colored differently than the receptor atoms. The authors should improve the representation of the complexes by hiding the aliphatic atoms.

A: We followed the reviewer's suggestions and changed the color of the ligand. We removed all aliphatic hydrogen atoms and even removed some less interacting amino acid residues.

5) Scheme 3. The authors should show the molecular structures and not cut'n'paste.

A: We followed the reviewer's suggestions and made appropriate changes by replacing former scheme 3 with new scheme 3.

6) MD trajectories (1 ns) are clearly too short to obtain relevant information, for analysis 1000 snapshots during 100 ns would be optimal.

A: In our MD simulations, we are interested in obtaining statistically relevant information regarding interaction energy, binding energy, and H-bonds. Although the literature mentions the cases with 100 ns https://www.ncbi.nlm.nih.gov/pmc/articles/PMC7534544/, it also states that even 10 ns would be enough. In addition, the same article (with 100 ns) states the following:

"We previously showed that several hundreds of picoseconds simulations could evaluate ligand binding stability in MDM2 better than AutoDock Vina scoring function (Ref 18 within the article). It is encouraging to notice that shorter simulations can effectively improve docking results because it can save resources, time, and be efficiently incorporated into a high-throughput workflow."

This citation is the reason why we believe that even 1 ns is enough for the MD step within our pipeline. Also, for assessing the binding energy the literature recommends only several ns. Articles: https://www.ncbi.nlm.nih.gov/pmc/articles/PMC2877349/ https://core.ac.uk/download/pdf/82762842.pdf

Other reviewers did request longer MD simulations. Still, we decided to follow the reviewer's request, in part, and carry the following additional MD simulations:  For Gromacs MD simulation and all the ligands mentioned in Table 3, interaction energy was recalculated for 10 ns (using 1000 snapshots). Additionally, for 9 ligands in Table 3, interaction energy was recalculated with 40 ns duration (using 1000 snapshots) and 100 ns duration (using 1000 snapshots), finally, for these 9 ligands, H-bond statistics were recalculated using 10 ns duration. Overall, we did not observe significant differences in interaction energy interaction energies between 1 ns, 10 ns, 40 ns, and 100 ns trajectories, (see Table SY), as well as no... significant differences in H-bond statistics (T-test between different groups representing MD duration attained no significant difference).

For Amber MM/PBSA calculations we also performed extended (5ns instead of 1ns) simulations for most of the samples. Again, no significant differences were observed except that binding affinity appears to be (on average) slightly higher in absolute value, both for the literature ligand set and selected lead compound set.

For some samples (five out of 30), the calculation broke and could not finish. Nevertheless, we decided to put all these additional and completed MD experiments and results into supplementary material of the manuscript (Larger Supplementary Table S6 and Table S9). We hope that these additional simulations and results represent a sufficient response to the reviewer's remark.

7) MM/PBSA, QM/MM, etc. are correct abbreviations for the methods.

A: We followed the reviewer's suggestion and made corresponding changes.

8) The pipeline is too complicated, I wonder if it makes sense to dock with GOLD and AD 4.2. Is the last step QM/MM necessary?

A: Although other reviewers do not necessarily see the pipeline as too complicated, we decided to do the following: 1) we removed "alternative pipeline (3x0.80)" from the pipeline and only gave a short note in the article with a separate note in Supplementary information note about alternative ways to filter compounds because alternative pipeline did not finally result in any new lead compound. We admit that no reviewer asked us to do that, but our workflow already contains two (slightly) different pipelines, one for the Drugbank dataset and the other for the COCONUT database. If we further insist on additional pipeline subroutes within one of the mentioned pipelines as "standard pipeline" and "alternative pipeline" this could only lead to the risk of confusing the broader readership. So we removed an alternative pipeline from our study, and also removed the term "standard pipeline". We hope that our decision is understandable. 2) We changed the pipeline in a way that we speak about the descriptor filtering section (which can be done with both GOLD and Adt4 using the rigid docking) and docking section (which can be done with flexible docking). We added few sentences explaining that rigid docking with Adt4.2 is not a must for the pipeline but can be also done with the GOLD program. 3) The last step QM/MM represents optimization and binding energy calculation of the optimal docked conformation. We find it important as it provides an improved binding score of the flexibly docked conformation. Many articles entail and describe QM/MM evaluations as contributions to better evaluation of binding affinity.

Therefore, this step is kept as part of the pipeline. However, we decided to provide the list of compounds prior to this last step and made corresponding changes in the manuscript. This list is provided in the supplementary as an illustration of the differences one can expect if the time-consuming and computationally demanding QM/MM optimizations are skipped in the pipeline.

We thank the reviewer for the constructive review.

Round 2

Reviewer 3 Report

Authors followed the recomendations. The article could be published in present form.